# Lottery Tickets on a Data Diet:
# Finding Initializations with Sparse Trainable Networks

**Mansheej Paul**[1][*]     **Brett W. Larsen**[1][*]
**Surya Ganguli**[1,2]     **Jonathan Frankle**[3,4,5]     **Gintare Karolina Dziugaite**[6,7]
[1]Stanford     [2]Meta AI     [3]MIT     [4]MosaicML     [5]Harvard     [6]Google Brain     [7]Mila; McGill

## Abstract

A striking observation about iterative magnitude pruning (IMP; Frankle et al. [10]) is that—after just a few hundred steps of dense training—the method can find a sparse sub-network that can be trained to the same accuracy as the dense network. However, the same does not hold at step 0, i.e., random initialization. In this work, we seek to understand how this early phase of pre-training leads to a good initialization for IMP both through the lens of the data distribution and the loss landscape geometry. Empirically we observe that, holding the number of pre-training iterations constant, training on a small fraction of (randomly chosen) data suffices to obtain an equally good initialization for IMP. We additionally observe that by pre-training only on "easy" training data we can decrease the number of steps necessary to find a good initialization for IMP compared to training on the full dataset or a randomly chosen subset. Finally, we identify novel properties of the loss landscape of dense networks that are predictive of IMP performance, showing in particular that more examples being linearly mode connected in the dense network correlates well with good initializations for IMP. Combined, these results provide new insight into the role played by the early phase training in IMP.

## 1 Introduction

Modern deep neural networks are often trained in the massively over-parameterized regime. Though these networks can eventually be pruned, quantized, or distilled into smaller networks, the resources required for the initial over-parameterized training poses a challenge to the democratization and sustainability of AI. This raises a fundamental question: under what circumstances can we efficiently train sparse networks? Recent work on the lottery ticket hypothesis [9, 10] has shown that, after just a few hundred steps of pre-training, a dense network contains a sparse sub-network that can be trained without any loss in performance. Finding this sparse sub-network currently requires multiple rounds of training to convergence, pruning, and rewinding to the pre-train point, a procedure termed iterative magnitude pruning (IMP, Figure 1; [9, 10]), Remarkably, even after all these rounds of training, we do not find trainable sparse sub-networks if we rewind to the random initialization; the first few hundred steps of dense network training is essential for finding sparse networks through IMP. In this work, we seek to understand this *very short but critical phase* of pre-training. In particular, we investigate the effect of training data and number of steps used during pre-training on the accuracy achieved by IMP. We also explain how certain properties of the loss landscape allow us to predict whether we will find a *matching initialization* (i.e., an initialization from which IMP "succeeds" in finding a subnetwork that can match the accuracy of the unpruned network, to be formalized later).

IMP proceeds in *three* phases: an initial *pre-training phase* where (1) the dense weights are trained for a few hundred steps, followed by (2) the resource-intensive *mask search phase*, during which we

---

[*]Equal contribution. Correspondence to: `{mansheej,bwlarsen}@stanford.edu; gkdz@google.com`

36th Conference on Neural Information Processing Systems (NeurIPS 2022).

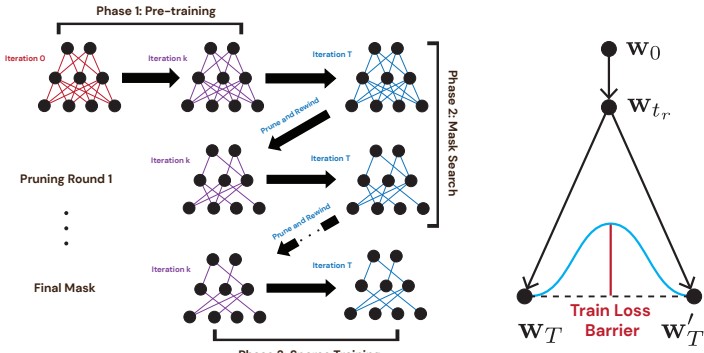

Figure 1: **Left:** Three phases of iterative magnitude pruning (IMP) with weight rewinding [10]. A dense network is trained in the pre-training phase for $t_r$ iterations, where $t_r$ is referred to as the rewinding iteration, and $\mathbf{w}_{t_r}$ is the parameters of the network at the rewinding point. The mask search phase produces a sparse subnetwork at a desired sparsity level by iteratively training, pruning the smallest magnitude weights, and rewinding to $\mathbf{w}_{t_r}$. The sparse training phase trains the final sparse subnetwork to convergence, starting with weights $\mathbf{w}_{t_r}$. **Right:** Illustration of computing the train loss barrier for initialization $\mathbf{w}_{t_1}$. $\mathbf{w}_T$ and $\mathbf{w}'_T$ are trained with different data order.

iteratively find a sparse mask by training the network and rewinding the weights repeatedly, finally, ending with (3) the *sparse training phase*, when the sparse masked network is trained. The weights learned after the pre-training phase—which serve as the initialization for the mask search phase—are referred to as the *pre-trained initialization* (see Figure 1).

In this work, we focus on studying the pre-training phase. To probe sufficient information needed to arrive at a matching initialization for IMP, we modify the pre-training phase by training with different pruned datasets and varying the number of pre-training steps; we then compare the performance of the resulting sub-networks across a range of sparsities. Following Paul et al. [23], we prune the datasets both randomly and according to EL2N scores, which estimate example difficulty by measuring early training performance in an ensemble of networks. We next turn to characterizing the loss landscape properties of matching initialization for IMP. Frankle et al. [10] investigate the relationship between successful initializations for IMP and linear mode connectivity on the *sparse* initialization for the entire dataset; here, we investigate this relationship on the *dense* initialization on a per-example basis. This enables us to identify signatures of the *dense* loss landscape correlated with a matching initialization. Thinking more broadly, we explore the relationship between sparse trainability and training stability. Gilmer et al. [13] demonstrate that, in hyperparameter regimes in which early training is unstable, learning rate warmup helps stabilize training. We build on this by investigating if the same pruned datasets which decreased the amount of pre-training required to find a matching initialization for IMP also decrease the required duration of learning rate warmup.

Overall, this empirical analysis provides new insight into the important and mysterious role that pre-training the dense network plays in IMP. Intriguingly, finding a matching initialization for IMP is an example of a problem in which the dominant paradigm of "more data, more training" is not optimal. Indeed, by carefully choosing the training examples, we can not only use a very small subset of the data but also reduce the training time required to solve this problem. From a scientific perspective, this suggests that different phases of training play different roles in the optimization process. Identifying and characterizing these phases will not only lead to a deeper understanding of the optimization of deep neural networks but also allow us to design training strategies that are optimal for each phase. From the practical standpoint, data loading is often an expensive part of the training process and can be a bottleneck especially early in training. By identifying the essential subset for this phase of training, our work may enable strategies to eliminate this bottleneck.

**Contributions.** We find empirical evidence for the following statements:

- **In the pre-training phase, only a small fraction of the data is required to find a matching initialization for IMP:** On standard benchmarks, across all sparsity levels we evaluated, we find that we can match accuracy by training on a small fraction of all of the available training data, selected randomly. (As we vary the amount of training data in pre-training phase, the number

of training iterations is held fixed.) Note that this observation changes if random label noise is introduced, in which case it becomes important to select easy (small EL2N score) examples.

- **The length of the pre-training phase can be reduced if we train only on the easiest examples:** Informally, training on a small subset of "easy-to-learn" training examples produces a better rewinding point than training on all data for the same number of iterations;

- **The quality of a pre-trained initialization for IMP correlates with more examples being linearly mode connected in the dense network:** This result complements the empirical evidence produced by Frankle et al. [10] connecting linear mode connectivity and the performance of IMP.

- **Training on easy data, which reduces the amount of pre-training required to find a matching initialization for IMP, does not reduce the amount of training required to stabilize the network via learning rate warmup ([13]):** The role played by data during IMP pre-training is thus different than that during learning rate warmup.

## 2 Background, Methods, and Related Work

We consider standard neural network training on image classification. Let $S = \{(\mathbf{x}_n, \mathbf{y}_n)\}_{n=1}^N$ denote training data, let $\mathbf{w}_t \in \mathbb{R}^D$ denote model parameters (weights) of the neural network, and let $\mathbf{w}_1, \mathbf{w}_2, \ldots$ be the iterates of (some variant) of SGD, minimizing the training *loss*, i.e., average cross-entropy loss over the training data. For a given training example $\mathbf{x}_n$, let $f(\mathbf{w}, \mathbf{x}) \in \mathbb{R}^K$ denote the logit outputs of the network for weights $\mathbf{w}$ and $p(\mathbf{w}, \mathbf{x}) = \sigma(f(\mathbf{w}, \mathbf{x}))$ be the probability vector returned by passing the logits through the softmax operation $\sigma$. By the *loss (error) landscape*, we mean the training loss (error), viewed as a function of the parameters. By training and test *error*, we mean the average 0-1 classification loss.

**Lottery ticket subnetworks.** The *lottery ticket hypothesis* [9] states that any standard neural network "contains [at initialization] a subnetwork that is initialized such that—when trained in isolation—it can match the test accuracy of the original network after training for at most the same number of iterations." Although such *matching subnetworks* (those that can train to completion on their own and reach full accuracy by following the same procedure as the unpruned network) are not known to exist in general at random initialization, they have been shown to exist after *pre-training* the dense network for a short amount of time (Phase 1 in Figure 1) before pruning [2, 10, 17, 25, 27].

Empirical evidence for this phenomenon comes via a procedure that finds such subnetworks retroactively after training the entire network. This procedure, called *Iterative Magnitude Pruning* [IMP; 10] is based on standard iterative pruning procedures [14], and can be decomposed into three phases (Figure 1), outlined in Algorithm 1.

---

**Algorithm 1** IMP rewinding to step $t_r$ and $N$ iterations.

---

1: Create a network with randomly initialization $\mathbf{w}_0 \in \mathbb{R}^d$.
2: Initialize pruning mask to $\mathbf{m} = 1^d$.
3: Train $\mathbf{w}_0$ for $t_r$ steps to $\mathbf{w}_{t_r}$                 ▷ Phase 1: Pre-Training
4: **for** $n \in \{1, \ldots, N\}$ **do**                          ▷ Phase 2: Mask Search
5:      Train the pruned network $\mathbf{m} \odot \mathbf{w}_{t_r}$ to completion. ($\odot$ is the element-wise product)
6:      Prune the lowest magnitude 20% of weights after training.
        Let $\mathbf{m}[i] = 0$ if the corresponding weight is pruned.
7: Train the final network $\mathbf{m} \odot \mathbf{w}_{t_r}$. Measure its accuracy.       ▷ Phase 3: Sparse Training

---

This procedure reveals the accuracy of pre-training the dense network for $t_r$ iterations, pruning, and training the pruned network thereafter. Phase 2 can be understood as an (expensive) oracle for choosing weights to prune at $t_r$. Although IMP is too expensive to use as a practical way to speed up training, it provides a window into a possible minimal number of parameters and operations necessary to successfully train a network to completion in practice. In our work, we extend this line of thinking, pursuing the minimal amount of data necessary to find and train these subnetworks. This is especially tantalizing due to the potential positive interactions between sparsity and minimizing the data necessary for training. *The result is a deeper inquiry into the minimal recipe for successful training and, thereby, into the fundamental nature of neural network learning in practice.*

In this respect, the closest work to ours is an experiment in a larger compendium by Frankle et al. [11] showing that the standard pre-training phase could be replaced by a much longer self-supervised phase.

Like our experiments in Section 3, that work aims to study what makes a matching initialization. Our approach and findings are substantially different, however: we reduce the number of examples rather than changing the labels, and we show that, not only are a small set of examples (starting at $\approx 2\%$) sufficient for pre-training, but also that they make it possible to pre-train in *fewer* steps.

There are many other ways to obtain pruned neural networks [e.g., 7, 14, 16, 20, 29]. The distinctive aspect of work on the lottery ticket hypothesis (and the one that makes it the right starting point for our inquiry) is that its goal is to uncover a minimal path from initialization to a trained network, regardless of the cost of doing so. The aforementioned procedures target real-world efficiency for training and/or inference.

**Linear Mode Connectivity and the Loss Barrier.** We investigate the error landscape using the parent–child methodology and instability analysis of Frankle et al. [10] and Fort et al. [8]. Writing $\mathrm{err}(\mathbf{w})$ for the test error at the weights $\mathbf{w}$, the (test) loss barrier between two networks $\mathbf{w}$ and $\mathbf{w}'$ is $\sup_{\alpha \in [0,1]}[\mathrm{err}(\alpha\,\mathbf{w} + (1-\alpha)\mathbf{w}') - (\alpha\,\mathrm{err}(\mathbf{w}) + (1-\alpha)\mathrm{err}(\mathbf{w}'))]$. This quantity measures the maximum increase in loss above the average loss along the linear path connecting the two networks on the loss landscape.

The loss barrier at iteration $t$ (with parent weights $\mathbf{w}_t$) is the loss barrier between the (children) weights $\mathbf{w}_T$ and $\mathbf{w}'_T$, where $\mathbf{w}_T$ and $\mathbf{w}'_T$ are copies of $\mathbf{w}_t$ trained to completion with the same procedure but different random seeds (minibatch order, GPU noise, data augmentation, etc.). See Figure 1 right for a visualization. Empirically, the loss barrier is achieved near $\alpha = \frac{1}{2}$, and so we compute it this way. The *onset of linear mode connectivity (LMC)* is defined to be the iteration $\bar{t}$ such that, for all $t > \bar{t}$, the error barrier at $t$ is zero. We follow Draxler et al. [4], Garipov et al. [12], and Frankle et al. [10] in considering the 0-1 loss barrier to be zero if it is less than 2%. Similar to Paul et al. [23], we also measure the error (0-1 loss) or cross-entropy loss barrier on individual training examples to explore the loss landscape corresponding to different subpopulations.

**Ranking training examples.** We define *"easy/hard data"* as the data that is ranked low/high, respectively, by the EL2N score introduced by Paul et al. [23]. EL2N scores depend on the margin early in training, and, loosely speaking, higher average margin early in training means lower importance for generalization of the final trained model. This connection to margin suggests that easy data is learned first (has higher margin early in training, maintained throughout the rest of training). EL2N scores were derived from the size of the loss gradient, and are thus are highly correlated with the magnitude of the gradient.

**Definition 2.1** (**EL2N Score**) *The EL2N score of a training sample $(\mathbf{x}, \mathbf{y})$ at iteration $t$ is defined as $\mathbb{E}\|p(\mathbf{w}_t, \mathbf{x}) - \mathbf{y}\|_2$, where the expectation is taken over $w_t$ conditioned on the training data.*

To calculate EL2N scores for a dataset, we follow the process outlined in [23]. In particular, we do the following:

1. Independently train $K = 10$ networks from different random initializations for $t$ iterations.

2. For each example and each network, we calculate the L2 norm of the error vector defined as $\|p(\mathbf{x}) - \mathbf{y}\|_2$ where $\mathbf{y}$ is the one-hot encoding of the label, and $p(\mathbf{x})$ are the softmax outputs of the network evaluated on example $\mathbf{x}$.

3. For each example, the EL2N score is the average of the error vector L2 norm across the $K$ networks.

In our experiments (Section 3), we vary the data that is accessible in the pre-training phase of IMP defined above. We either choose the data that we feed to the algorithm at random while preserving class balance, or based on the EL2N scores.

**Stabilizing early training via learning rate warmup.** Learning rate warmup period can be seen as a form of pre-training, allowing one to eventually train at higher learning rates and larger batch sizes. Training with large batches and high learning rates is desirable in practice, as it allows for more efficient GPU utilization and may reduce the total number of updates needed to achieve the desired accuracy. Gilmer et al. [13] empirically observe that learning rate warmup essentially improves the optimization by allowing the initial optimization trajectory to navigate to "flatter" optimization landscape, i.e., one with smaller highest loss Hessian eigenvalue.

# 3 The Role of Training Data Selection in Pre-Training

As can be seen in Figure 2, when training sparse networks using IMP with rewind step $t_r = 0$, the final test accuracy of the sparse networks falls off rapidly with increasing sparsity. However, as we increase the rewind step $t_r$, the network performance improves across all sparsity levels and at a rewind step $t^*$, the network performs as well as or better than the dense network at high sparsities. Informally, training the dense network for $t^*$ steps creates a matching initialization for IMP. But what does the network learn in these first $t^*$ steps? In this section, we take the first step towards answering this question by investigating which subsets of the training data are sufficient for finding a matching initialization. In order to compare networks trained on different subsets of data for different numbers of iterations, we introduce the notion of a *matching initialization* with the following definitions.

**Definition 3.1** *Let $\mathbf{w}_t^S$ be the dense network weights after training on a subset of the training data, $S$ until rewind step $t$. Then for two data subsets $\{S, S'\}$, rewind times $\{t, t'\}$ and a given range of sparsities, $\mathbf{w}_{t'}^{S'}$ is said to* dominate *(weakly dominate) $\mathbf{w}_t^S$ if sparse networks obtained from IMP with $\mathbf{w}_{t'}^{S'}$ as the initialization achieve better (no-worse) accuracy than those obtained from IMP with $\mathbf{w}_t^S$ as the initialization.*

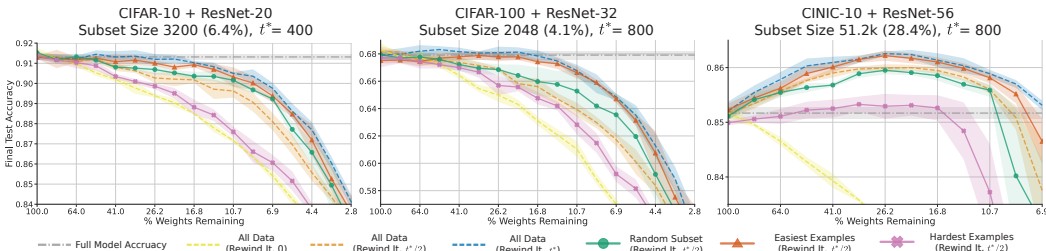

Figure 2: For a given rewind step $t_r = t^*/2$, training on a small fraction of random data during the pre-training phase of IMP leads to matching initializations (compare the solid green with circles and dashed orange curves) across dataset, network, and hyperparameter configurations. Using just the easiest training examples during this phase produces a matching initialization for rewind point $t^*$ in just $t^*/2$ steps (compare the solid red with triangles and dashed blue curves). Pre-training on the hardest examples is detrimental to the performance of the initialialization (solid pink curve with crosses). IMP with rewinding to initialization (dashed yellow curve) and the dense model (dashed grey curve) are used as baselines. For each dataset + network configuration, we present the best performing easy data subset size. For a sweep across subset sizes, see Figure 3 and the Appendix E.

For a network trained on the full dataset for $t$ steps, we write $\mathbf{w}_t$. In Figure 2, we see that $\mathbf{w}_{t^*}$ dominates $\mathbf{w}_{t^*/2}$ which in turn dominates $w_0$. We investigate which data subsets $S$ and rewind steps $t$ lead to networks $\mathbf{w}_t^S$ that dominate $\mathbf{w}_{t^*}$ and $\mathbf{w}_{t^*/2}$—such networks are called matching initializations.

**Definition 3.2** *A dense network $\mathbf{w}_t^S$ is a matching initialization for rewind time $t^*$ if $\mathbf{w}_t^S$ weakly dominates $\mathbf{w}_{t^*}$.* [2]

We empirically find that certain surprisingly small subsets $S$ and rewind step $t_r < t^*$ lead to matching initializations for rewind time $t^*$.

**Experimental design.** To evaluate the effect of the training subset size and composition on the quality of the pre-trained initialization, we train ResNet-20/ResNet-32/ResNet-56 on subsets of CIFAR-10/CIFAR-100/CINIC-10, respectively. The subset size $M$ is varied and subsets are chosen as follows: (i) $M$ randomly selected examples, distributed equally among all classes; (ii) the easiest $M$ examples; (iii) the hardest $M$ examples. The easiest examples are those with the smallest EL2N scores and the hardest are the examples with the largest EL2N scores [23].

---

[2]Note that due to the stochasticity in training and risk measurements, we ignore small deviations in the final test accuracy. Thus one rewinding point could weakly dominate another one even if at some sparsity levels their mean performance "crosses" over while approximately remaining within the standard error of one another.

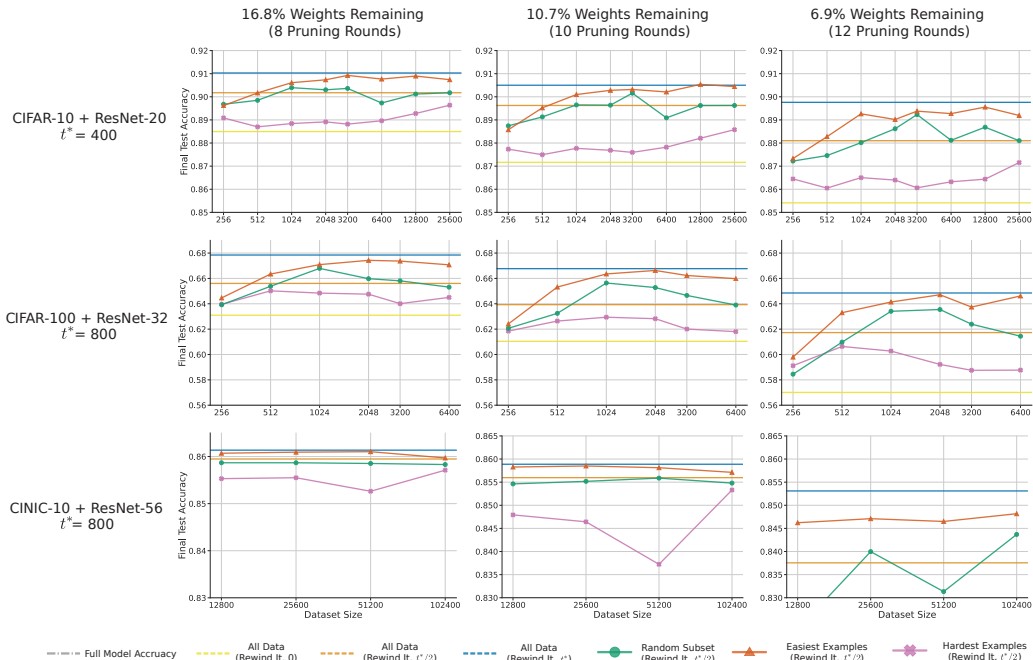

Figure 3: A summary of the the dependence on subset size for the style of experiments described in Figure 2. The first column represents the performance across subset size for the fixed sparsity 16.8% weights remaining or 8 rounds of pruning. The subsequent columns show the same for 10.7% (10 pruning rounds) and 6.9% (12 pruning rounds) respectively. The horizontal lines correspond to baseline runs at rewind steps 0, $t^*/2$, and $t^*$ using all the data. For CINIC-10 (bottom row), rewind step 0 and the hardest data subsets are not visible in some cases because their accuracies fall below the range displayed.

Due to the significant computational demands of performing IMP with multiple pre-training schemes and replicates, we focus on a targeted set of pre-training iterations $t_r$. In particular, we study the pre-training iteration $t_r = t^*$ where training on all examples leads IMP to find spare sub-networks that perform as well as the dense network for a large range of sparsities ($t^* = 400$ for CIFAR-10 and 800 for CIFAR-100 and CINIC-10). We also study the more challenging pre-training iteration of $t_r = \frac{t^*}{2}$, where pre-training on all data does not yield a matching initialization. When $M$ examples are not enough to train for $t_r$ iterations without replacement, we make multiple passes over the $M$ examples as necessary. Figure 2 shows the best performing easy data subset for each dataset across the full range of sparsities; Figure 3 shows the performance across subset size at three fixed sparsities.

**Randomly chosen examples.** Pre-training the dense networks on small, randomly chosen subsets $S$ can lead to initializations for IMP, $\mathbf{w}_{t_r}^S$, that dominate initializations $\mathbf{w}_{t_r}$, trained on the entire training set for the same number of steps. In Figure 2 we see that for all dataset + network combinations, pre-training the dense network on a small random subset (solid green curve with circles; sizes ranging from 4.1% for CIFAR-100 to 28.4% for CINIC-10) for $t_r = t^*/2$ leads to initializations that (weakly) dominate those that were obtained from training the network for the same number of steps on all the data. This observation leads to a surprising suggestion: in these experiments, the subset size is smaller than the total number of images seen during the pre-training phase; for the particular goal of finding a matching initialization of IMP, multiple passes through the same small dataset can be more beneficial than seeing more random data.

**Easiest examples (lowest EL2N scores).** By pre-training on just the easiest examples (identified by lowest EL2N scores, solid red curve with triangles in Figure 2), we can obtain matching initializations in fewer steps compared to training on the full dataset. In Figure 2, we see that for all three dataset and network combinations and for the subset sizes shown, the initialization obtained from training on the easiest examples for $t_r = t^*/2$ steps leads to matching initializations for $t^*$.

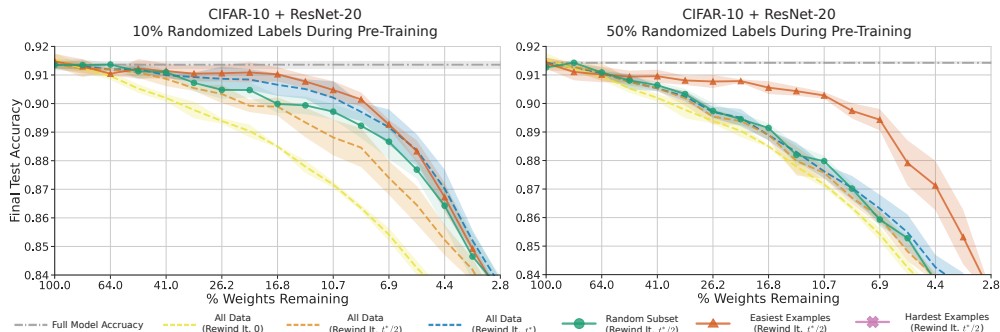

Figure 4: Pre-training on a random subset or all data is not robust to label noise during this initial phase of IMP. However, pre-training with the easiest data as scored by EL2N scores computed from the corrupted dataset is robust. In both the left (10% randomized labels) and right (50% randomized labels), pre-training on the easiest data for $t^*/2$ iterations dominates all other pre-training schemes, including training on all data for $t^*$ iterations. Results for additional subset sizes are included in Appendix E.

**Hardest examples (highest EL2N scores).** Conversely, pre-training on the hardest examples (solid pink curve with crosses in Figure 2) yields worse accuracies than pre-training on all examples or a random subset. In fact, on CIFAR-10 the hardest examples perform little better than using no pre-training at all. Interestingly, when training a dense network, these hardest examples are crucial for obtaining a network with good generalization properties [23]. This suggests that while the hard example may be key later in training, repeated passes through easier examples should be the focus during the very early stages of training to quickly find a good initialization for IMP.

**Randomized labels during pre-training.** Pre-training on a all data or a random subset is not robust to corruption with random label noise [28] during the pre-training phase. As seen in Figure 4, the higher the percentage of randomized labels, the lower the performance of these data subsets, and in particular, the pre-trained rewinding point becomes no better than a random initialization when $50\%$ of the labels are randomized. On the other hand, training on easiest data with EL2N scores computed on the corrupted dataset is robust to this noise (solid red curve with triangles in Figure 4). This is because examples with randomized labels are hard ([23]) according to this metric, and thus the easiest examples will select a subset of largely uncorrupted data.

**Summary.** Taken together, our results suggest that, finding a matching initialization for IMP at rewinding step $t^*$ is an interesting problem in which "more data, more training" is *not* optimal; it is neither necessary to train on all the data nor to train for the full $t^*$ steps. In fact, we can get away with training on a surprisingly small dataset for as little as half the number of steps if we make multiple passes through the *right* examples, here the easiest examples as defined by lowest EL2N scores.

## 4 Pre-training through the lens of Linear Mode Connectivity

Frankle et al. [10] observed a strong indicator of whether a *sparse* sub-network generated by IMP would be able to match the accuracy of the dense network is whether or not that sparse sub-network train to the same linearly connected mode (i.e. the train loss barrier is close to 0, see Figure 1). Here we ask what properties of the *dense* network at the rewind point might also be predictive of this property. In the architectures + datasets we consider, the onset of linear mode connectivity (LMC) occurs in the dense network later than the first rewind time which produces a good initialization for IMP (i.e. $\bar{t} > t_r$), and thus, we consider two generalizations to the notion of LMC. First, we look at the train loss barrier on a per-example basis meaning that instead of considering the full loss landscape we separately look at the performance of the linear interpolation on each example (the original notion of train loss barrier is obtained by averaging these values). Second, we then look at the distribution of these per-example train loss barriers as a continuous measure of the state of the network rather than simply considering whether their average is 0. We demonstrate in general that these measures of the loss landscape of the dense network correlate well with the IMP performance of the pre-trained initialization. We additionally show that pre-training on easy data results in a

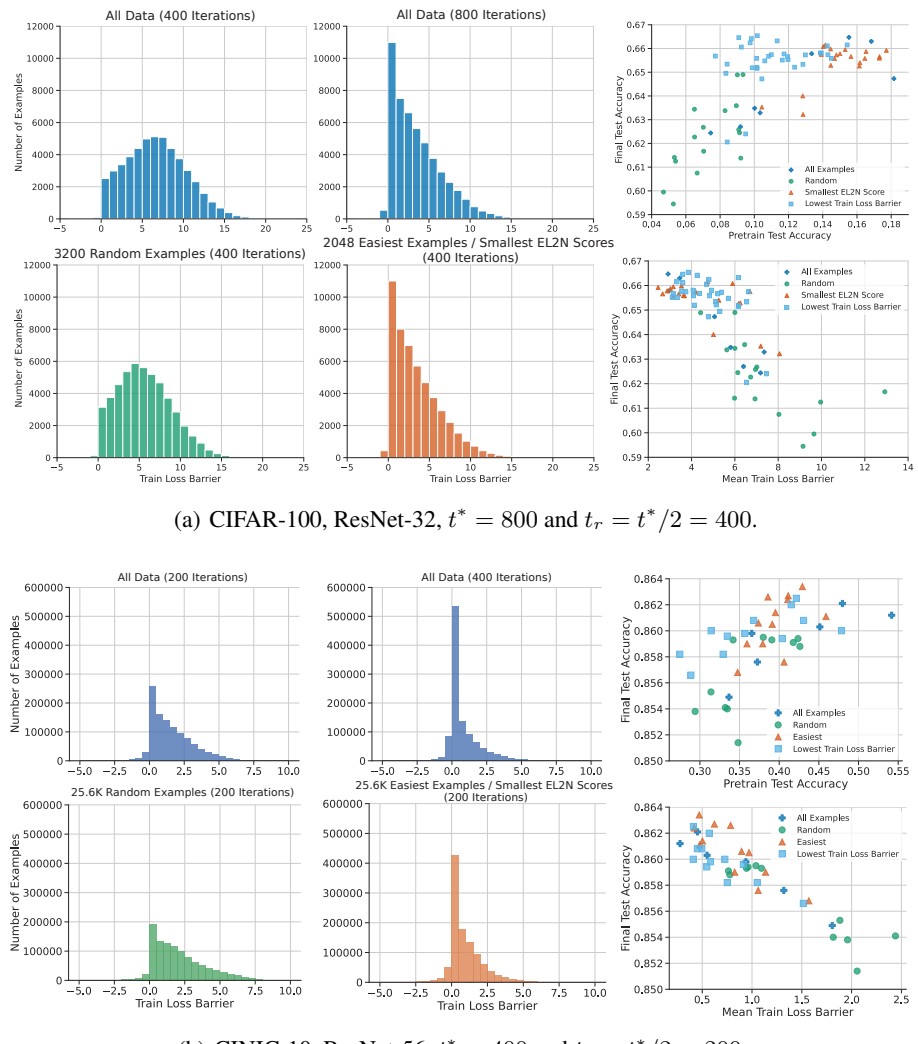

(a) CIFAR-100, ResNet-32, $t^* = 800$ and $t_r = t^*/2 = 400$.

(b) CINIC-10, ResNet-56, $t^* = 400$ and $t_r = t^*/2 = 200$.

Figure 5: **Left and center columns:** Pre-training on all the data for $t^*$ (top right) vs. $t^*/2$ (top left) steps leads to a distribution shift to smaller per-examples train loss barriers between children runs of the dense network. When pre-trained for just $t^*/2$ iterations but with a random subset of data, the distribution is similar to pre-training on all data for $t^*/2$ steps. However, when pre-trained on a subset of easy examples for $t^*/2$ iterations, the distribution of train loss barriers displays a shift comparable to pre-training on all the data for $t^*$ iterations. As seen in Figure 2 and the Appendix, pre-training with this random subset matches pre-training with all data for $t^*/2$ steps while pre-training with the easiest data for $t^*/2$ matches pre-training on all data for $t^*$ steps. This observation suggests that the per-example distribution of train loss barriers may be a useful signature for predicting the IMP performance of dense initializations. **Right column:** Scatter plots of final test accuracy vs. pre-train accuracy of the dense initialization (top) and the mean train loss barrier (bottom) across a variety of dataset pruning strategies for pre-training. Final test accuracy is at 8.6% weights remaining (11 pruning rounds) for CIFAR-100 and 21% weights remaining (7 pruning rounds) for CINIC-10. We observe that the mean train loss barrier is better correlated with the final test accuracy than the pre-train accuracy of the dense initialization.

distribution with a large fraction of examples linearly mode connected faster than pre-training on all data. See Appendices B and C for additional experiments and an exploration of why this occurs.

**The distribution of per-example loss barriers.** In the left and center columns of Figure 5 we plot histograms of per-example loss barriers for various rewinding points and data subsets for CIFAR-100 (panel a) and CINIC-10 (panel b). There is a clear shift in the empirical distribution of the per-example loss barriers — not only the mean is shifting as previously observed, but also the mode. Pre-training

on all data for $t^*$ iterations has the same effect on the distribution of the loss barriers as pre-training on the easy examples for $t^*/2$ iterations, two procedures which produce matching IMP initializations.

Given the connection between the error barrier and IMP sub-network performance, it is natural to ask whether examples which achieve low loss barrier early in training with the full dataset are more important for producing a matching initialization than high loss barrier examples. We call this ranking of the data by the per-example loss barrier at a given iteration the LMC score, and the blue squares in the scatter plot are the result of training on a subset of the data determined by this score. Our empirical findings suggest that using this score does not match the performance of using easy data (see Appendix D).

**The correlation between final test accuracy and mean train loss barrier.** The right column of Figure 5 shows a scatter plot of final test accuracy of the sparse trained network vs. two different properties of the dense initialization: test accuracy of this pre-train point (top) and the average train loss barrier from 3 pairs of children runs (bottom). Here we observe the same result across the different training conditions: a correlation between the train loss barrier when spawning from $w_{t_r}$, and the final test accuracy of the sparse sub-networks. The correlation is weaker for the pre-train test accuracy, suggesting that the loss landscape properties hold more import for determining the success of IMP. Furthermore, we see that training on the easiest examples produces rewinding points with smaller train loss barriers than training on random subsets.

## 5 The Role of Training Data in Learning Rate Warmup

To this point, we have studied the early phase of training through the lens of the lottery ticket hypothesis. However, the dynamics of the early phase of training have important implications beyond the sparse regime considered by lottery ticket research. For example, learning rate warmup is a nearly ubiquitous part of training state-of-the-art models. Warmup can be viewed as an accommodation for the idiosyncrasies of the earliest part of training and a pre-training strategy that prepares the network to train at the full learning rate. In this section, we extend our previous experiments to dense training with learning rate warmup by ablating the training data during this phase to assess whether easy data alone suffices, similar to our observations in the pre-training phase for IMP (Section 3).

**Easy data does not reduce the amount of training required to stabilize the network via learning rate warmup.** The effects of performing learning rate warmup with easy, random, and hardest subsets of the data appear in Figure 6. Intriguingly, the most striking performance drop is observed when doing learning rate warmup with easy data, especially at longer pre-training times. Pre-training on the random and hardest data produces the same performance for sufficiently large subset sizes. The role played by data during IMP pre-training is thus different than that during learning rate warmup; the easy data does not add stability benefits during the later. We hypothesize that this is because learning rate warmup happens over a much larger fraction of the total training time than IMP pre-training.

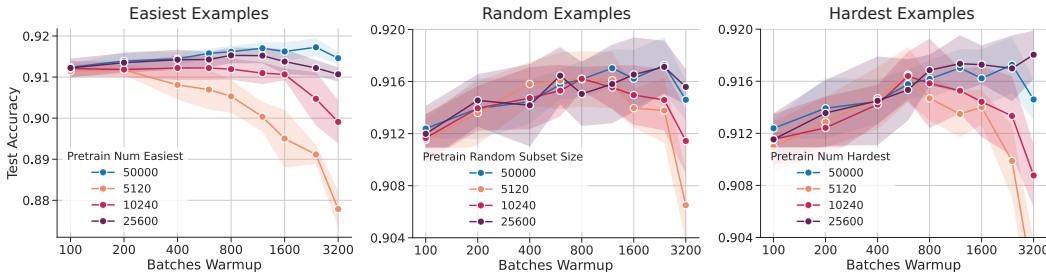

Figure 6: Training with different data subset sizes and pruning strategies for the learning rate warmup period for ResNet-20 on CIFAR-10. In the large batch size and learning rate regime, learning rate warmup is essential for stabilizing training; we sweep over different learning rate warmup periods with a batch size of 1024. In contrast to pre-training for IMP, training on easiest examples in this initial phase leads to worse performance than training with all data, random subsets, or the hardest data. We note that the learning rate warmup period is a much longer fraction of training than IMP pre-training which lasts for only a few hundred steps.

**Discussion and algorithmic implications.** The benefits of training on easy data—faster learning and increased stability to SGD noise—are only present in the very early phase of learning. Paul et al. [23] have shown that, if we continue to train on easy data only, the network will converge to a lower final test accuracy. So a natural question is what other scenarios do the benefits of training on easy data extend to? We began to investigate this question here by studying training at high learning rates where learning can become unstable early in training and is usually stabilized by learning rate warmup [13]. Our results show that training only on easy examples does not help and in fact hurts performance in this case. Two possible reasons for this behavior are (1) learning rate warmup is often used for the first 10% of training, which may be too long for easy examples to be helpful (the pre-training phase in iterative magnitude pruning where easy examples are clearly beneficial can be as little as 2% of training) and (2) the instability in this case—loss diverging to infinity—is not reduced by methods that improve stability to SGD noise. Despite the negative results of this experiment, we think that, when combined with our previous results, the core scientific findings in our work are interesting and broadly useful to the community. In particular, some algorithmic implications are as follows: (1) While developing algorithms for training sparse networks or finding matching networks at initialization, researchers might want to incorporate methods or search for initializations that improve stability to SGD noise. (2) Recent work in curriculum learning found that curricula provide marginal benefits unless there are additional constraints such as a training budget [26]. Our work helps narrow the time window in which training on easy examples is beneficial and provides another example in which, for the constrained scenario of finding matching sparse networks, a simple curriculum of training on easy data first is beneficial. Thus, these findings may help in the design of better training curricula. (3) In the training of large language models, a common problem is that loss spikes and training becomes unstable. A hypothesis for why this phenomenon occurs is that the network is presented with a bad batch or a set of bad batches that destabilize training. Since this is related to instability due to SGD noise (bad batches are randomly sampled) it may be possible to improve training stability by injecting easy examples into the training process.

# 6 Discussion

Recent empirical evidence has shown that deep neural network optimization proceeds in several distinct phases of training [8, 11]. Understanding the role that data plays in these different phases can help us characterize what is being learned during them. Since data loading is often a bottleneck, this understanding also has the potential enable more efficient training schemes. In this work, we have considered two essential phases of early training: pre-training for IMP which enables sparse optimization and learning rate warmup which stabilizes network training. Our experiments identify what data is sufficient for each of these procedures, and as they both occur early in training, these finding can be used to design more efficient data loaders for streaming datasets. Furthermore, as IMP is a computationally expensive procedure, understanding how it works is essential for designing better algorithms with the same performance. To this end, we identify loss landscape properties of the dense network initialization for IMP that are predictive of successful spare training. Though this work does not provide an improved algorithm for obtaining sparse networks, we believe our results provide essential guidance for researchers pursuing algorithms that perform pruning early in training (i.e. finding sparse masks without training to convergence).

# Acknowledgements

The experiments for this paper were funded by Google Cloud research credits. S.G. thanks the James S. McDonnell and Simons Foundations, NTT Research, and an NSF CAREER Award for support while at Stanford. This work was done in part while G.K.D. was visiting the Simons Institute for the Theory of Computing. The authors would like to thank Daniel M. Roy for feedback on multiple drafts.

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
