# A Experimental Details

**Code.** The code used to run the experiments is available at: `https://github.com/mansheej/lth_diet`

**Datasets.** We used CIFAR-10, CIFAR-100 [18], and CINIC-10 [3] in our experiments. For CINIC-10, we combine the training and validation sets into a single training set with 180,000 images. The standard test set of 90,000 images is used for testing. Each dataset is normalized by its per channel mean and standard deviation over the training set. All datasets get the same data augmentation: pad by 4 pixels on all sides, random crop to $32\times32$ pixels, and left-right flip image with probability half.

**Models.** In these experiments we use ResNet-20, ResNet-32, ResNet-56 [15]. These are the low-resolution CIFAR variants of ResNets from the original paper. The variants of the network used are specified in the figures.

**Randomized Labels.** The labels were randomized during the pre-training phase *only* by first selecting 10%/50% of the training uniformly at random and then drawing a new label uniformly from the 10 classes of CIFAR-10. Note that because this procedure can result in an example being reassinged the correct label, on average only 9%/45% of the labels are corrupted by this procedure. The dataset is corrupted once and then resused across all subset sizes and replicates.

The EL2N scores used to determine the subset of easiest data are computed by training on the corrupted dataset. As seen in Figure 5 of [1], the network typically learns the correct label early in training for corrupted data points. As a result, the corrupted examples will be ranked as difficult by the EL2N scores as verified in [23]. Thus, the noisy labels are filtered out by pre-training on the examples with the lowest EL2N scores.

**Linear mode connectivity procedure.** To determine if two children of a given network are linearly mode connected, we perform the following procedure:

1. Spawn two children initialized with the current weights of the given network.
2. Independently train the two networks with different SGD noise realized through different data orders.
3. Calculate the mid-point in weight space between the two networks.
4. Calculate the training loss for the resulting network at the mid-point.
5. Calculate the training loss barrier—the loss calculated at the mid-point minus the mean of the losses calculated at the endpoints.

Train loss barriers are calculated both on a per-example basis and as averages over the whole dataset. The per-example training loss barrier is just the training loss barrier evaluated on that example. In practice, we train three children and calculate the train loss barrier between each pair and average. Following [10], we say that two networks are linearly mode connected if the train loss barrier between them is less than 2%.

**EL2N score computation** To calculate EL2N scores for a dataset, we follow the process outlined in [23]. In particular, we do the following:

1. Independently train $K = 10$ networks from different random initializations for $t$ iterations.
2. For each example and each network, we calculate the L2 norm of the error vector defined as $\|p(\mathbf{x}) - \mathbf{y}\|_2$ where $\mathbf{y}$ is the one-hot encoding of the label, and $p(\mathbf{x})$ are the softmax outputs of the network evaluated on example $\mathbf{x}$.
3. For each example, the EL2N score is the average of the error vector L2 norm across the $K$ networks.

To calculate these scores, we use ResNet-20 and $t = 7800$ iterations for CIFAR-10, ResNet-32 and $t = 7800$ iterations for CIFAR-100, and ResNet-56 and $t = 8000$ iterations for CINIC-10.

**Hyperparameters.** Networks were trained with stochastic gradient descent (SGD). $t^*$ was chosen for each dataset such that $t_r = t^*$ produces sparse networks that can be trained to the same accuracy as the dense network to at least 16.8% sparsity (8 pruning levels). The pre-training learning rate was chosen based on which of the set $\{0.1, 0.2, 0.4\}$ produced the best performance at $t^*$. The full hyperparameters are provided in Table 1.

Table 1: Hyperparameters Used for Experiments.

|  | CIFAR-10 | CIFAR-100 | CINIC-10 |
|---|---|---|---|
| ResNet Variant | ResNet-20 | ResNet-32 | ResNet-56 |
| Batch Size | 128 | 128 | 256 |
| Pre-training Learning Rate | 0.4 | 0.4 | 0.1 |
| Learning Rate | 0.1 | 0.1 | 0.1 |
| Momentum | 0.9 | 0.9 | 0.9 |
| Weight Decay | 0.0001 | 0.0001 | 0.0001 |
| Learning Rate Decay Factor | 0.1 | 0.1 | 0.1 |
| Learning Rate Decay Milestones | 31200, 46800 | 31200, 46800 | 15625, 23440 |
| Total Training Iterations | 62400 | 62400 | 31250 |
| IMP Weight Pruning Fraction | 20% | 20% | 20% |

All results are reported as the mean and standard deviation of 4 replicates. The only exception is Figure 5 scatter plots where each point corresponds to 1 replicate.

**Learning rate warmup experimental design.** For the learning rate warm-up experiments, we train ResNet-20 on CIFAR-10 with batch size = 1024. For the optimizer, we use SGD with learning rate = 3.2, momentum = 0.9 and weight decay = 0.0001. There is an initial linear learning rate warm-up for $t$ batches. The network is then trained for a total of 160 epochs with a LR drop by a factor of 10 at 80 and 120 epochs. In Figure 6, we sweep the learning rate warmup period $t$. For each hyperparameter configuration, we report the mean and standard deviation of 8 replicates. In addition to all the training data, we perform experiments with different subsets of the training data with sizes 5120, 10240 and 25600. In these cases, the network was trained on the data subset during the warm-up period only and on the full dataset afterwards. For the easiest (hardest) examples, we choose examples with the lowest (highest) EL2N scores which were calculated as described above.

**Compute Resources.** The experiments were performed on virtual Google Cloud instances configured with 4 NVIDIA Tesla A100 GPUs. Each experiment replicate was run on a single A100 GPU. The approximate compute time for a full run of IMP was 8 hours for CIFAR-10, 12 hours for CIFAR-100, and 10 hours for CINIC-10.

**Ethical and societal consequences.** Being empirically-driven work, the experiments performed consumed considerable energy. However, we view our work as a step towards the long-term goal of improving the efficiency of training neural network which will ideally lead to an increase in the democratization and sustainability of AI. Towards this aim, our paper provides insight into the role of data in early training and a better understanding of which properties of pre-trained initializations enable sparse optimization. We hope this will inspire better algorithms for training sparse networks.

## B Explanation of the Role of Easy Data

Our primary observation in Section 3 is that training on a small fraction of easy data can reduce the number of steps needed to find an initialization which contains a matching sparse network, i.e. one that can be trained to the same accuracy as the dense network. An explanation for this phenomenon can be divided into two questions. First, what characteristic of a pre-trained dense network can predict if it contains a matching sparse network? Second, does training only on easy examples allow us to find networks with this characteristic faster and why?

In Section 4, we try to provide a linear mode connectivity (LMC) based explanation for the first question. Previous work [10] has shown that the rewind step at which sparse networks become matching correlates with the onset of LMC for the sparse network. Intuitively, this suggests that a notion of stability is important for finding matching sparse networks—such networks exist at the

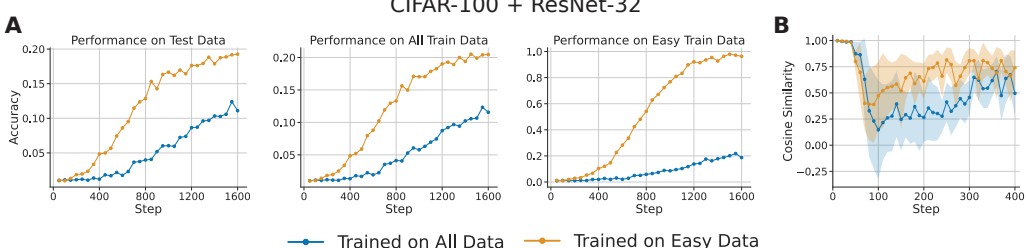

Figure 7: (a) Comparison of the performance of a network trained on all data and easy data during the first 1600 steps of training. Training on easy data results in superior test accuracy as well as train accuracy on both the full dataset and the easy subset. (b) Comparison of gradient stability for a network trained on all data and easy data. Every 10 iterations, we sample 16 batches and plot the cosine similarity between every batch's gradient and the mean gradient across batches. We see that this measure of similarity is consistently higher when trained on easy data.

rewind step at which the sparse network optimization becomes stable to SGD noise. However, this occurs before the onset of LMC in the dense network. Thus, we generalized the notion of stability to SGD noise from a binary metric—is the train loss barrier below some threshold—to a continuous one—the average train loss barrier—and found that this characteristic of the dense network is indeed correlated with our ability to find matching initializations.

To answer the second question, we show that training only on easy examples finds networks with lower average train loss barriers faster and the resulting distribution of per-example train loss barriers is very similar to those obtained from training on all data for twice as many steps (Section 4, Figure 5). Here we further investigate why this might be the case. First, we compare how fast networks learn when trained on easy vs. all data. For CIFAR-100 + ResNet-32, we train for 1600 steps on both a subset of 2048 easy examples and on all examples. We measure the test accuracy as well as the train accuracy on both the full dataset and the easy subset. The results in Figure 7a show that training on easy examples performs better on all 3 metrics, leading to faster learning. Next we compare the coherence of gradients along these training trajectories. Every 10 steps, we sample 16 batches from the dataset used to train the network and calculate the mini-batch gradients. For each step, we then measure the cosine similarity between each sampled gradient and the mean gradient across batches. The results in Figure 7b show that this similarity is consistently higher when training on easy data.

Previous work has shown that networks learn easy examples before learning hard examples [21]. Our results further show that, very early in training, making repeated passes through the easy data creates a stronger and more coherent training signal which leads to faster learning. An intuitive summary of these results is, given that the information necessary for finding matching subnetworks is learned in the first few hundred steps, it is more efficient to focus training resources on examples that can be learned well early in training.

## C   Other Hypotheses for the Role of Easy Data: Negative Results

Pre-training on easy data (lowest EL2N scores) allows us to reduce $t_r$ without hurting test accuracy of the sparse networks found by IMP. One possible reason for this is that networks pre-trained on just the easy data achieve better test accuracies at step $t_r$. But in Figure 5, we find that the accuracy of the dense network at the pre-trained initialization does not completely explain the improved performance. Here we explore two additional hypotheses we considered for what role the easy data plays during the pre-training phase that we found to be not explanatory.

### C.1   Changes in the gradient size.

1. Train a ResNet-20 on CIFAR-10 for 200 iterations on a random subset of 25600 examples. At every iteration record the gradient norm of the batch.

2. Repeat the above process but training on a subset of 25600 examples with the smallest EL2N scores.

3. Compare the distribution of gradient norms

Examples with lower EL2N scores have smaller gradient norms at the point at which they are computed as observed by [23]. We investigate whether the gradient norm is also smaller when training on the easiest examples during the pre-training phase of IMP, which would reproduce the effects of gradient clipping which is often beneficial early in training. We compare the distribution of batch gradient norms seen in the first 200 steps while training on easy data versus a random subset of data. Note that since we are only training for 200 steps with a batch size of 128, training on all the data is equivalent to training on random data. We run the following experiment:

Figure 8 shows the results. We find that, when training on easy examples, the batch gradient norms are typically larger. Our hypothesis is that on easy examples, the gradients are more aligned, leading to larger average gradients. We thus reject the hypothesis that pre-training on the easy subset of the data effectively performs gradient clipping.

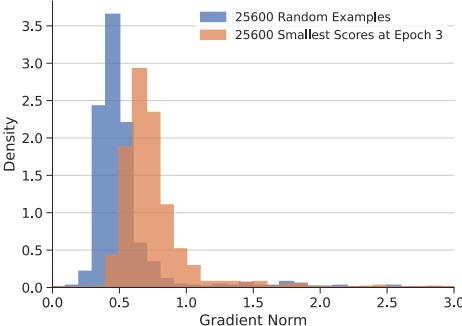

Figure 8: The histogram of minibatch gradient norms during the first $t_r = 200$ iterations of training (CIFAR-10, ResNet-20), when the minibatches are sampled from a random subset of 25600 examples compared to sampled from a subset of the 25600 easiest examples.

## C.2 Change in distribution of per-example error

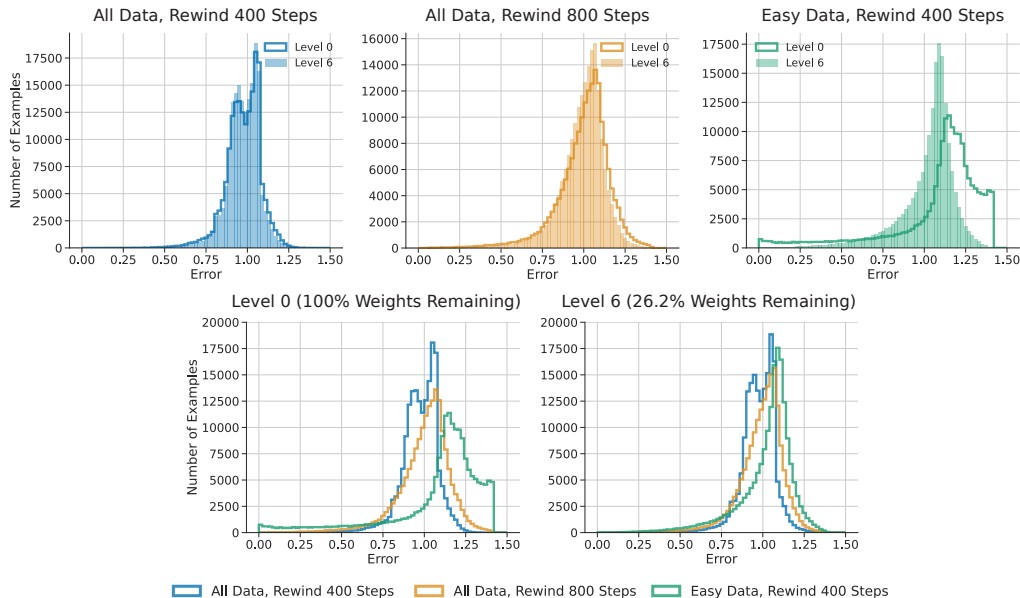

Figure 9: The distribution of per-example error at the pre-training point $t_r$ for several different pre-training procedures before and after sparse projection. These errors are calculated for a single ResNet-56 network on CINIC-10 (i.e. not averaged across replicates). The top row shows the distributions grouped by pre-training procedure while the bottom row shows all error distributions in the dense networks (left) and sparse networks (right).

For a ResNet-56 trained on CINIC-10, we considered whether different pre-training procedures resulted in diffferent distributions of the per-example error early in training. We made the following observations in the results shown in Figure 9

1. Prior to the sparse projection, there is a distinctive distribution of error after training on easy data. A split occurs in the data where a portion of the examples shifts towards zero error while the remainder increases in error as compared to training on all data (level 0 green curve vs. level 0 blue curve in Figure 9).

2. When comparing the dense vs. sparse networks pre-trained on all data, the distribution looks the same before and after projection (level 0 vs. level 6 blue and orange curves). However, for pre-training on the easy data a significant change happens after the projection (level 0 vs. level 6 green curves).

3. After the projection, the error distributions from training on all data for 800 iterations and easy data for 400 iterations are similar (level 6 orange and green curve) while the training on all data for 400 iterations is different (level 6 blue curve).

These observations bear similarity to the per-example train loss barrier results considered in the main text (Figure 5). Pre-training on easy data for 400 iterations produces results that have the same signatures as pre-training on all data for 800 iterations (and differ from training on all data for 400 iterations). However, the per-example train loss barrier provides a clearer explanation and thus we think is the more fundamental quantity.

## D  Linear Mode Connetivity: Additional Experiments

Here we present additional work on understanding the connection between linear mode connectivity of a dense initialization and its performance as a pre-trained initialization for IMP. We introduce a new score for ranking examples in terms of the per-example train loss barrier early in training which we call the LMC score. We then compare this score to EL2N and test the IMP performance of pre-training on the subset of examples with low LMC scores.

### D.1  Background on (linear) mode connectivity.

The deep learning phenomena of *mode connectivity* via simple paths has been uncovered by Draxler et al. [5], Garipov et al. [12]. In particular, it was shown that SGD-trained deep neural network modes are connected via simple nonlinear paths, along which the training and test loss is approximately constant. Since then this phenomena has also been studies theoretically [19]. A related mode property has been described in the literature on generalization in deep learning: Nagarajan and Kolter [22] show that for small networks trained on MNIST there exist *linearly* connected modes. Surprisingly, a slightly modified version of this phenomena is also true for large scale vision networks: Frankle et al. [10] show that coupling the first training epochs restricts SGD to converge to the same linearly connected mode. More recently, Entezari et al. [6] hypothesize that linearly connected modes are equivalent up to symmetries. Their empirical observations suggest that there is overlap between the functions parameterized by different modes.

### D.2  LMC Scores

Our observation that successful pre-train initializations are correlated with a shift to smaller per-example train loss barriers (left and center columns of Figure 5) naturally raises the question: are examples that become linearly mode connected first also the ones which are important for this phase of training? To this end, we introduced the notion of a per-example LMC score which allows us to rank examples in this manner:

**Definition D.1** *The* LMC score *of an example is the loss barrier computed on a dense network at rewinding iteration $t$ averaged over $K$ instances.*

Under this definition, the train error barrier is the LMC scores averaged over the training data. If we compute the score after the onset of linear mode connectivity $\bar{t}$ it will not provide an informative ranking of the data as the score will be close to 0 for all examples; we compute the scores before $\bar{t}$.

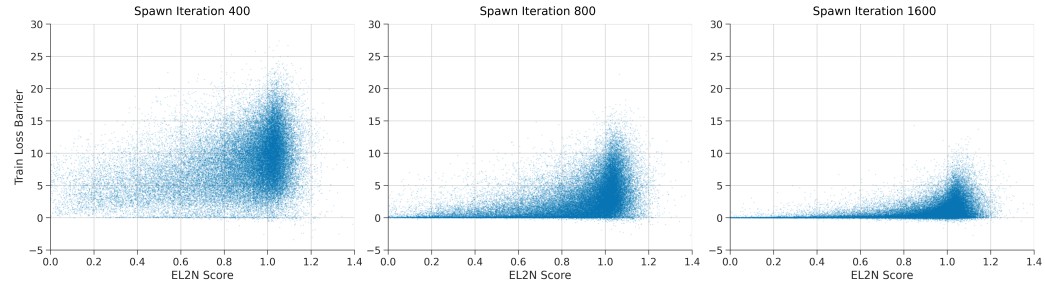

(a) CIFAR-100, ResNet-32, Batch Size 128.

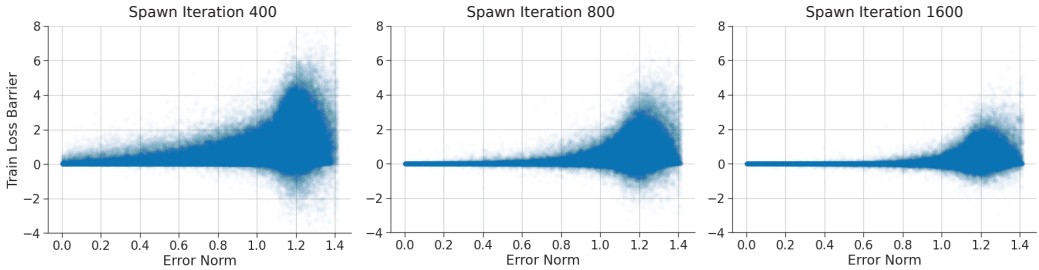

(b) CINIC-10, ResNet-56, Batch Size 256.

Figure 10: Scatter plot of the relationship between the EL2N score computed at 1600 iterations and the Linear Mode Connectivity (LMC) score computed at iteration 400, 800, and 1600 iterations.

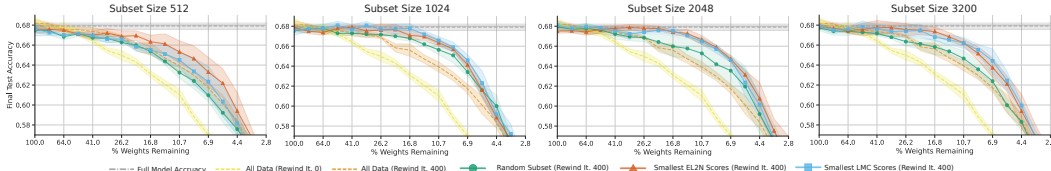

(a) CIFAR-100, ResNet-32, $t^*/2 = 400$ LMC scores computed after 1600 iterations (batch size 128).

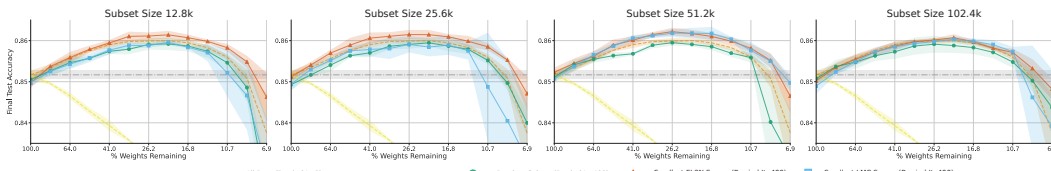

(b) CINIC-10, ResNet-56, $t^*/2 = 400$. LMC scores computed after 800 iterations (batch size 256).

Figure 11: Performance of IMP pre-training using data subsets determined by smallest LMC scores (solid light blue curve with squares). This is compared against random subsets (solid green curve with circles) and easiest examples (solid red curve with triangles). Pre-training is performed with learning rate 0.4 and batch size 128 for CIFAR-100 and learning rate 0.1 and batch size 256 for CINIC-10. Several of these runs are included in the scatters plot in the right column of Figure 5 with the method labelled as "Lowest Train Loss Barrier."

In our experiments, we computed the LMC score by averaging over three children runs from two replicates, so that each replicate provided three train loss barriers (one for each pair of children) and the total average was taken over six instances.

We visualize the correlation between EL2N scores and LMC scores. The scatter plots in Figure Figure 10 show that low EL2N score examples tend to be the ones for which the LMC score goes

down rapidly with spawning time, and is nearly-zero early in training. However, the opposite is not true, and high EL2N scores have a wide range of per-example loss barriers.

### D.3 Pre-training on Low LMC Scores

Figure 11 shows that rewinding points obtained by training on low LMC score examples dominate random example pre-training, but are (weakly) dominated by training on low EL2N score examples. We thus conclude that low EL2N examples are more representative of the necessary and sufficient subset of examples needed to find a good rewinding point.

## E   Full Results

### E.1   Comparison to Forgetting Scores

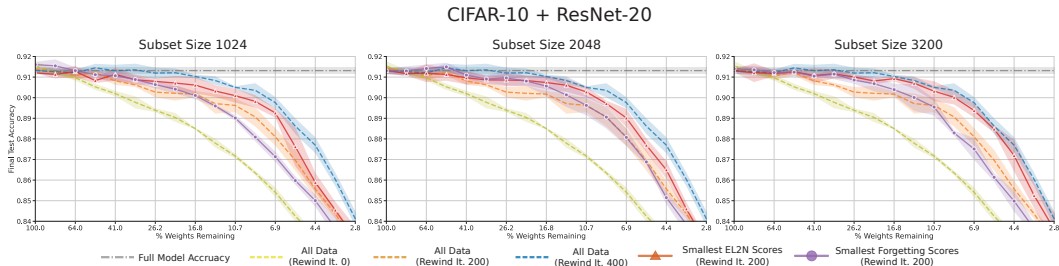

Figure 12: Comparison of pre-training on smallest EL2N scores and forgetting scores on CIFAR-10 + ResNet-20. On this dataset, we observe superior performance of EL2N scores.

In Figure 12, we compare pre-training on the smallest EL2N scores vs. smallest forgetting scores. The forgetting score of an example is defined by Toneva et al. [24] as the number of times the example is included in a mini-batch and the network switches from a correct to incorrect classification of that example—a "forgetting event". As seen in the results, pre-training on the subset of smallest forgetting scores underperforms using the subset of smallest EL2N scores computed early in training for subset sizes 1024, 2048, and 3200.

There are two reasons why we might expect forgetting scores to underperform EL2N scores. First, forgetting scores as defined here look at example difficulty over the whole course of training whereas we computed the EL2N scores early in training at 16 epochs. For IMP pre-training, we are most interested in finding examples which the network finds easy early in training. Second, for many of the easy examples the forgetting score is 0 meaning the example is learned once and never forgotten. If this is true for a large number of examples, this ranking is less useful when trying to choose a small subset size (for both scores, the subset was class-balanced).

Our goal was to identify training data subsets that suffice for pre-training. Our choice of metric was thus intentional: EL2N scores capture how difficult it is to learn an example early in training, which intuitively correlates to stability of training and error barriers.

### E.2   Results Across Subset Size

Here we present the full set of experiments performed for the results in the main text, sweeping across subset size. Figure 13 and Figure 14 show the results for performing IMP pre-training with different data subsets across a range of different subset sizes. Figure 15 shows the result for performing IMP pre-training with the dataset corrupted by randomized label noise (the original dataset is then used for the mask search and sparse training phase).

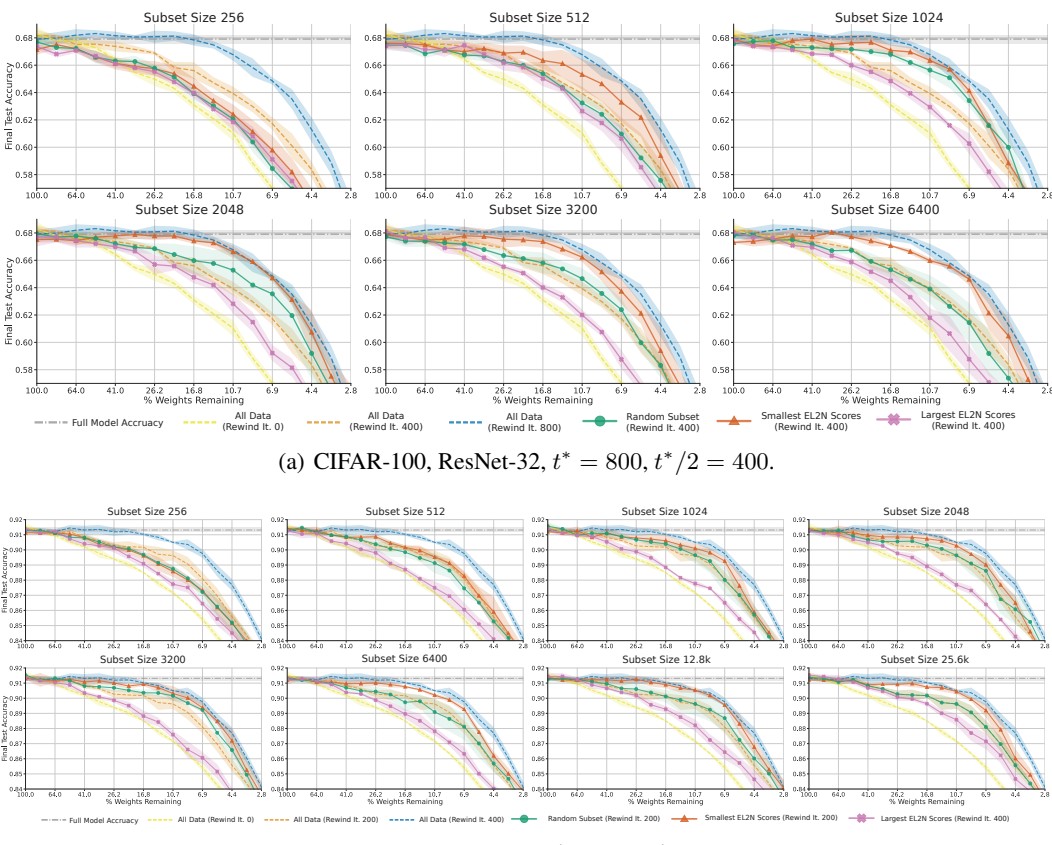

(a) CIFAR-100, ResNet-32, $t^* = 800, t^*/2 = 400$.

(b) CIFAR-10, ResNet-20, $t^* = 400, t^*/2 = 200$.

Figure 13: Extended results for Figure 2 and Figure 3 across all subset sizes considered on CIFAR-10 and CIFAR-100. For each dataset, pretraining was performed with all data for $t^*/2$ steps (dashed orange curve) and $t^*$ steps (dashed blue curve). Performing IMP as an initialization is included as a baseline (dashed yellow curve). IMP pre-training was then performed with three different data subsets for $t^*/2$ steps: random examples (solid green curve with circles), easiest examples (solid red curve with triangles), and hardest examples (solid pink curve with crosses). For both CIFAR-100 and CIFAR-10, a learning rate of 0.4 and batch size 128 was used during the pre-training period.

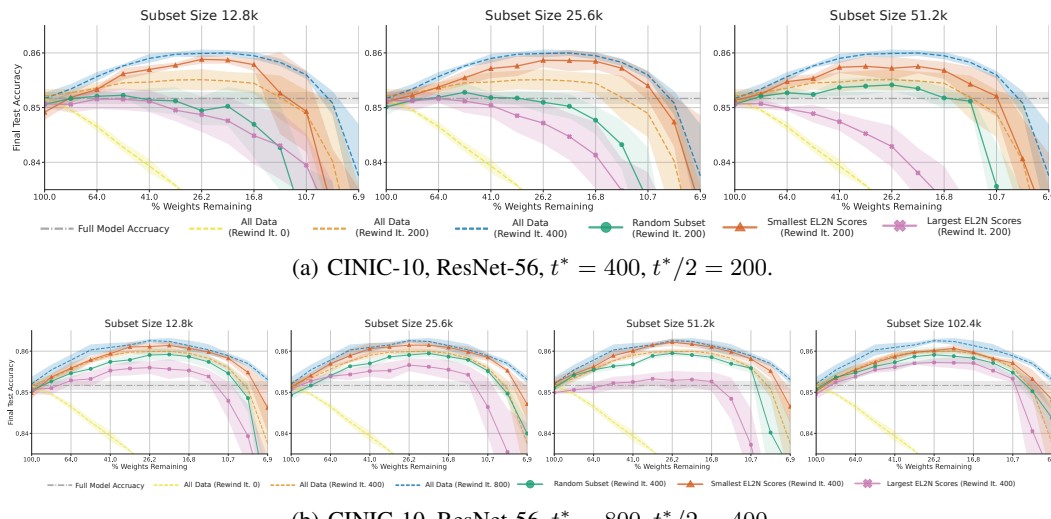

(a) CINIC-10, ResNet-56, $t^* = 400$, $t^*/2 = 200$.

(b) CINIC-10, ResNet-56, $t^* = 800$, $t^*/2 = 400$.

Figure 14: Extended results for Figure 2 and Figure 3 across all subset sizes considered on CINIC-10. For each dataset, pretraining was performed with all data for $t^*/2$ steps (dashed orange curve) and $t^*$ steps (dashed blue curve). Performing IMP as an initialization is included as a baseline (dashed yellow curve). IMP pre-training was then performed with three different data subsets for $t^*/2$ steps: random examples (solid green curve with circles), easiest examples (solid red curve with triangles), and hardest examples (solid pink curve with crosses). For CINIC-10, a learning rate of 0.1 and batch size 256 was used during the pre-training period.

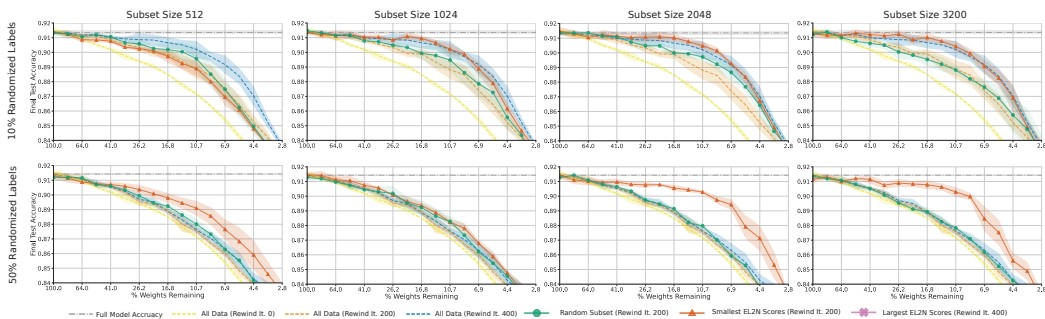

Figure 15: Extended results for Figure 4. In the top row, pre-training is performed on the dataset with 10% randomly corrupted labels; in the bottom row, pre-training is performed with 50% randomly corrupted labels. The corruption is performed once and then is held the same across subset sizes and replicates. Here $t^* = 400$, and the random and easy data subsets are trained for $t^*/2 = 200$ steps. Pre-training was performed with a learning rate of 0.4 and batch size of 128.