# OpenReview forum: "Lottery Tickets on a Data Diet: Finding Initializations with Sparse Trainable Networks"
_NeurIPS.cc/2022/Conference — NeurIPS 2022 Accept_

### Official Review · Reviewer_pW7h · 2022-07-10

**Rating:** 6
**Confidence:** 3
**Soundness:** 3 good
**Presentation:** 3 good
**Contribution:** 2 fair

**Summary:**

This paper studies the role of data in finding early-stage lottery tickets. With carefully-designed experiments, the authors showed that training on a small fraction of easy data suffices to get a good initialization for IMP and in particular it takes fewer pre-train iterations. In addition, the authors proposed to use the distribution of per-example loss barriers to predicting the quality of IMP initialization. Overall, these findings offer interesting insights into the early phase neural network training dynamics.

**Questions:**

See my comments in the weaknesses section.

Other questions and suggestions:
- For experiments with only a subset of data, do you also use the same subset of data for IMP or the whole dataset?
- I encourage the authors to dig a little deeper into why easy data would help. I believe this would make the paper much stronger.

**Limitations:**

The authors discussed the limitations at the end of the paper. I don't see any negative societal impact of this work.

**Strengths And Weaknesses:**

Strengths:
- The paper is well-written and quite easy to follow.
- Although the ingredients are not new, the idea of studying the role of data in finding good lottery tickets is novel. I believe some of the findings of this work are interesting to many people in our community.
- Experiments are well-designed and support the authors' claims very well.

Weaknesses:
- All experiments are done on relatively small datasets. Quite often, the results on CIFAR do not transfer to larger datasets like ImageNet. One or two runs on ImageNet would make the results more convincing and I believe nowadays most academic labs can afford to run a few experiments on ImageNet.
- The authors only used one single metric to rank the training data. It would be great if the authors could try out one or two other metrics. In addition, it seems that the data selection procedure could be expensive. I think the authors should add some discussions in the main paper about that.
- The authors discovered an interesting phenomenon but didn't give any explanation of why easy data could help.

---

> ### Author Response · Authors · 2022-08-02
> **Response to Reviewer pW7h 1/2**
>
> We thank the reviewer for their enthusiasm and helpful comments about clarifying the role of easy data in pre-training and the reason for using EL2N as opposed to other scores.  Below we address these concerns.
>
> **Explanation of why easy data helps.**  Our primary observation is that training on a small fraction of easy data can reduce the number of steps needed to find an initialization which contains a matching sparse network, i.e. one that can be trained to the same accuracy as the dense network. An explanation for this phenomenon can be divided into two questions. First, what characteristic of a pre-trained dense network can predict if it contains a matching sparse network? Second, does training only on easy examples allow us to find networks with this characteristic faster and why?
>
> In Section 4: we try to provide a linear mode connectivity (LMC) based explanation for the first question. [Previous work](https://arxiv.org/abs/1912.05671) has shown that the rewind step at which sparse networks become matching correlates with the onset of LMC for the *sparse* network. Intuitively, this suggests that a notion of stability is important for finding matching sparse networks—such networks exist at the rewind step at which the *sparse* network optimization becomes stable to SGD noise. However, this occurs before the onset of LMC in the *dense* network. Thus, we generalized the notion of stability to SGD noise from a binary metric—is the train loss barrier below some threshold—to a continuous one—the average train loss barrier—and found that this characteristic of the *dense* network is indeed correlated with our ability to find matching initializations.
>
> To answer the second question, we showed that training only on easy examples finds networks with lower average train loss barriers faster and the resulting distribution of per-example train loss barriers is very similar to those obtained from training on all data for twice as many steps (Section 4, Figure 5). In this revision (Appendix B), we add a new set of experiments to further investigate why this might be the case. First we compare how fast networks learn when trained on easy vs all data. For CIFAR-100 + ResNet-32, we train for 1600 steps on both a subset of 2048 easy examples and on all examples. We measure the test accuracy as well as the train accuracy on both the full dataset and the easy subset. The results in Figure 7a show that training on easy examples performs better on all 3 metrics, leading to faster learning. Next we compare the coherence of gradients along these training trajectories. Every 10 steps, we sample 16 batches from the dataset used to train the network and calculate the gradients on those batches. For each step, we then measure the cosine similarity between each sampled gradient and the mean gradient across batches. The results in Figure 7b show that this similarity is consistently higher when training on easy data.
>
> [Previous results](https://openreview.net/forum?id=HkxHv4rn24) have shown that networks learn easy examples before learning hard examples. Our results further show that, very early in training, making repeated passes through the easy data creates a stronger and more coherent training signal which leads to faster learning. An intuitive summary of our results is, given that the information necessary for finding matching subnetworks is learned in the first few hundred steps, it is more efficient to focus training resources on examples that can be learned well early in training.

---

> ### Author Response · Authors · 2022-08-02
> **Response to Reviewer pW7h 2/2**
>
> **Comparison to Other Scores.**  In our revision, we have added a comparison to forgetting scores for CIFAR-10.  The forgetting score of an example is defined by [previous work](https://openreview.net/pdf?id=BJlxm30cKm) as the number of times the example is included in a mini-batch and the network switches from a correct to incorrect classification of that example—a “forgetting event”.  As seen in the results in Figure 12, pre-training on the subset of smallest forgetting scores underperforms using the subset of smallest EL2N scores computed early in training for subset sizes 1024, 2048, and 3200.  The original draft of the paper also already included comparison to LMC scores which we introduce and study in Appendix D to capture which examples contribute to the error barrier and instability of training.
>
> There are two reasons why we might expect forgetting scores to underperform EL2N scores.  First, forgetting scores as defined here look at example difficulty over the whole course of training whereas we computed the EL2N scores early in training at 16 epochs.  For IMP pre-training, we are most interested in finding examples which the network finds easy early in training.  Second, for many of the easy examples the forgetting score is 0 meaning the example is learned once and never forgotten.  If this is true for a large number of examples, this ranking is less useful when trying to choose a small subset size (for both scores, the subset was class-balanced).
>
> Our goal was to identify training data subsets that suffice for pre-training. Our choice of metric was thus intentional: EL2N scores capture how difficult it is to learn an example early in training, which intuitively correlates to stability of training and error barriers.
>
> **Extension to ImageNet.**  In the time allotted for revision, we were not able to finish experiments on ImageNet.  For each baseline and data subset, a full training run is required for each pruning level of IMP such that one replicate of these experiments requires a full day running on 8 A100 GPUs.  We did, however, make progress on these experiments and successfully ran the baseline experiments with the full datasets which can be found [here](https://anonymous.4open.science/r/36EB/IMPWR_IMAGENET_RN50-TestAcc_Density.png).  We agree that examining how these results extend to ImageNet is an interesting question and we will aim to include these experiments in the camera-ready version if accepted.
>
> >For experiments with only a subset of data, do you also use the same subset of data for IMP or the whole dataset?
>
> After the dense pre-training phase, the whole dataset is used for the remainder of the IMP algorithm.
>
> **Summary of changes in this revision:**
> * Explanation for why easy data helps find initializations containing matching subnetworks clarified and additional experiments added in Appendix B.
> * Additional motivation for learning rate warmup and algorithmic implications added in Appendix E.
> * Additional experiment with different scores for finding easy data added in Appendix F.1.
> * Currently running additional experiments on different architectures and datasets (VGG16 on CIFAR10) and (ResNet50 on ImageNet). These will be included in the camera ready version if accepted. Intermediate results (baseline runs) at: https://anonymous.4open.science/r/36EB/README.md

---

### Official Review · Reviewer_exXP · 2022-07-11

**Rating:** 7
**Confidence:** 4
**Soundness:** 3 good
**Presentation:** 4 excellent
**Contribution:** 3 good

**Summary:**

This paper focuses on the pre-training phase of the IMP procedure. The authors show that only a small fraction of data is required for finding a matching initialization. In particular, the authors show that the length of such pre-training is reduced if we use as training set only the easiest examples. Then, they also introduce two measures of the loss landscape of the dense network that correlate well with the IMP performance of the pre-trained initialization. The authors also investigate if the same behavior can be observed during learning rate warmup.

**Questions:**

The authors present all the results using the ResNet architecture. It would be interesting to show if the same behavior is confirmed also with other architectures (such as VGG or WideResNet).  I understand that a thorough validation with different architectures will require a significant additional number of experiments. However, I think that it is still interesting to verify the most important assumptions with other architectures in a limited number of experiments.
I found the experiment with randomized labels (pag. 7) not very fair because, when you perform training on easiest data, you select almost only data with uncorrupted labels.

**Limitations:**

The authors adequately addressed the limitations and potential negative societal impact of their work


**Strengths And Weaknesses:**

The paper provides new insights into the role of the pretraining phase for finding matching lottery tickets. I think that the empirical evidence shown in this paper can significantly contribute to the research in this field, reducing the amount of data and time required for this training phase. The contributions presented in this paper also help to shed light into the behavior of the network during such pretraining phase.
The paper is well written and easy to follow. The experimental validation of the assumptions presented in the paper is solid.

---

> ### Author Response · Authors · 2022-08-02
> **Response to Reviewer exXP**
>
> Thank you for your encouraging comments and enthusiasm for our results. We hope that you will continue to champion this work. We address your questions below.
>
> > It would be interesting to show if the same behavior is confirmed also with other architectures (such as VGG or WideResNet).
>
> We agree that this is worth checking. We are currently running additional experiments to confirm that our main result—the length of pre-training required for finding matching sparse networks using iterative magnitude pruning (IMP) can be reduced by training on easy data only—holds in the case of VGG-16 trained on CIFAR-10. However, since these experiments involve multiple training runs for each run of IMP and sweeps over both rewinding step and pre-training data subset size, we are unable to complete them in the time allocated for rebuttals.  We did make progress on these experiments and successfully ran the baseline experiments with the full datasets which can be found [here](https://anonymous.4open.science/r/36EB/CIFAR10-VGG.png). If accepted, we will aim to include the full results in the camera ready version.
>
> > I found the experiment with randomized labels (pag. 7) not very fair because, when you perform training on easiest data, you select almost only data with uncorrupted labels.
>
> We absolutely agree with the reviewer; pretraining on easy data is so effective because we almost only select data with uncorrupted labels. But our goal with this experiment wasn’t to demonstrate the superiority of pre-training on easy examples. Rather, we tried to provide context for the surprising success of pretraining on a tiny subset of random examples. For example, with CIFAR-10 + ResNet-20, pretraining on 3200 random examples for 200 steps leads to better performance than on all the data for 400 steps! We hypothesized that this is because, on CIFAR-10 in particular, most of the examples are relatively easy and a random subsample is mostly composed of these easy examples. In a dataset with a larger fraction of hard examples, selecting easy examples should more significantly outperform a random subset. By adding label noise, we make the subset of randomly selected examples harder and, as expected, performance of pre-training on the random subset decreases while that of pre-training on easy data remains high. We view the robustness of easy examples to label noise as a secondary benefit.
>
> **Summary of changes in this revision:**
> * Explanation for why easy data helps find initializations containing matching subnetworks clarified and additional experiments added in Appendix B.
> * Additional motivation for learning rate warmup and algorithmic implications added in Appendix E.
> * Additional experiment with different scores for finding easy data added in Appendix F.1.
> * Currently running additional experiments on different architectures and datasets (VGG16 on CIFAR10) and (ResNet50 on ImageNet). These will be added to the camera ready version if accepted. Intermediate results (baseline runs) at: https://anonymous.4open.science/r/36EB/README.md

---

### Official Review · Reviewer_jz9s · 2022-07-17

**Rating:** 9
**Confidence:** 4
**Soundness:** 4 excellent
**Presentation:** 4 excellent
**Contribution:** 4 excellent

**Summary:**

The paper deals with performing pruning with respect to data. The emphasis of this work is in understanding the dynamics of the learning process. More specifically how do the early stages of the training help you with performing pruning based on the neural network weights.

**Questions:**

Did you try this on different modalities of data ?
How does the loss landscape look like before and after the pruning ? Did you try and visualize it ?

**Limitations:**

No There is no discussion on it (or perhaps I missed it in the supplementary material )

**Strengths And Weaknesses:**

The paper is extremely clearly written. The claims are very crisp and clear. More importantly the impact of this is pretty huge, if this will hold at scale for very large data sets.
The method section is well explained and the (continually hammered in ) connection to the lottery ticket hypotheses gives good context to the solution
Albeit there are a few things that are still very empirical/ad hoc:
line 136 - is the 2% a stiff criteria ? Or did you try to change it. More importantly what would have been very convincing if this would have been tried on several modalities of data. I have a sneaking suspicion that for tabular/time series this would not hold.
Line 147 - Is P the probability of the network, the wights ?

A few typos and assortment of other things
line 115, missing a space
line 118 - double dashes
* sometimes authors relay way too much on references for example EL2N context some of the benchmarks of GWMP other methods etc...
** This is extremely nitpicking but, the reviewer is colorblind, such it took me a but to correlate in Fig 5 Easiest examples and low EL2N, can you please make it consistent between the title of the figures and legend of the scatter plots ?

---

> ### Author Response · Authors · 2022-08-02
> **Response to Reviewer jz9s**
>
> Thank you for your enthusiasm and encouraging comments. We hope that you will continue to champion this work. We address your questions below.
>
> > Albeit there are a few things that are still very empirical/ad hoc: line 136 - is the 2% a stiff criteria ? Or did you try to change it?
>
> While this a now-standard part of the definition (motivated originally by a calculation of the Monte Carlo noise in the test error estimate itself), this work does not use the 2% criteria because we define a continuous measure: the distribution of per-example loss barriers.  This was included to give context for the usual definition of LMC.
>
> > [Strengths and Weaknesses] More importantly what would have been very convincing if this would have been tried on several modalities of data. I have a sneaking suspicion that for tabular/time series this would not hold. [Questions] Did you try this on different modalities of data ?
>
> In line with standard practices in the neural network pruning literature and in the data pruning literature, we focused on computer vision tasks. This made it possible for us to most effectively build on and compare to prior work.  While we agree that additional modalities would allow us to make more general claims, our experiments convincingly support the claims we make as they pertain to the central benchmarks of study in the literature we are contributing to.
>
> > Line 147 - Is P the probability of the network, the weights ?
>
> P is the softmax output of the network.
>
> > A few typos and assortment of other things line 115, missing a space line 118 - double dashes.
>
> Thank you, these typos have been corrected.
>
> >Sometimes authors rely way too much on references, for example EL2N, some of the benchmarks of IMP, other methods, etc…
>
> We have provided a more thorough explanation of these topics in the appendix.  For example, Appendix A includes the complete procedure for computing EL2N scores and Linear Mode Connectivity as well as the hyperparameters used to produce the IMP baselines.
>
> >This is extremely nitpicking but the reviewer is colorblind, such that it took me a bit to correlate in Fig 5 Easiest examples and Low EL2N. Can you please make it consistent between the title of the figures and legend of the scatter plots ?
>
> The titles of the plots have been changed in Fig. 5 to say “Easiest Examples / Smallest EL2N Scores” to make the connection to the scatter plot clearer.
>
> > How does the loss landscape look like before and after the pruning ? Did you try and visualize it ?
>
> Though we don’t visualize the loss landscape before and after pruning, in line with [previous](https://arxiv.org/abs/1912.05671) [work](https://arxiv.org/abs/2107.07075) we used linear mode connectivity (LMC) to probe the structure of the loss landscape.  LMC provides a measure of how stable the loss landscape is to SGD noise.  [Frankle et al.](https://arxiv.org/abs/1912.05671) studied the network at the rewind point *after pruning* and showed that this notion of stability is important for finding matching subnetworks. However, the rewind step where we begin to find matching subnetworks occurs before the onset of LMC in the *dense* network.  We thus extended this analysis to the network at the rewind point *before pruning* by generalizing the notion of stability to SGD noise from a binary metric—is the train loss barrier below some threshold—to a continuous one—the average train loss barrier.  We found that this characteristic of the dense network is indeed correlated with our ability to find matching initializations.
>
> **Summary of changes in this revision:**
> * Explanation for why easy data helps find initializations containing matching subnetworks clarified and additional experiments added in Appendix B.
> * Additional motivation for learning rate warmup and algorithmic implications added in Appendix E.
> * Additional experiment with different scores for finding easy data added in Appendix F.1.
> * Currently running additional experiments on different architectures and datasets (VGG16 on CIFAR10) and (ResNet50 on ImageNet). These will be added to the camera ready version if accepted. Intermediate results (baseline runs) at: https://anonymous.4open.science/r/36EB/README.md

---

> > ### Comment · Reviewer_jz9s · 2022-08-07
> > **Response to authors**
> >
> > Thank you so much for the corrections and for the references.
> > I think the new appendix E actually really sheds light on some technicalities I am missing.
> > A follow up to some of the answers given.
> >
> > Re modality of data, I know that everyone is doing just vision but if everyone keeps on doing it we won't really understand just what is going on in extremely high noise to signal ratio data sets. Or in regimes where you are data starved
> >
> > > While this a now-standard part of the definition (motivated originally by a calculation of the Monte Carlo noise in the test error estimate itself), this work does not use the 2% criteria because we define a continuous measure: the distribution of per-example loss barriers. This was included to give context for the usual definition of LMC.
> > > P is the softmax output of the network.
> > If accepted and you have the extra page i would love a small paragraph stating this a bit more explicitly.
> >
> > Likewise regarding compute assessment that you have in the supp mat. If accepted I would be happy if this could be moved into the main text. I believe in all pruning work, this is super important to give context of resources used

---

> > > ### Author Response · Authors · 2022-08-09
> > > **Response to Reviewer jz9s**
> > >
> > > Thank you for the feedback.  If accepted, we will use the extra space to move the material recommended by the reviewer from the appendix to the main text for added clarity.

---

### Official Review · Reviewer_xZ2E · 2022-07-17

**Rating:** 6
**Confidence:** 4
**Soundness:** 3 good
**Presentation:** 3 good
**Contribution:** 3 good

**Summary:**

This paper studies iterative magnitude pruning (IMP), which is used to find a sparse sub-network that can train to achieve the accuracy dense networks can achieve. The authors found that empirically:
- Training on a small fraction of randomly chosen data is sufficient to find a good initialization
- Training on easy training data can reduce the number of steps to find a good initialization
- Mean train loss barrier is predictive of final test accuracy.



**Questions:**

- B.1 refutes the hypothesis that training on easy data is doing gradient clipping. Are there any other hypotheses?
- How are the results in setion 5 related to finding good sparse initialization? Does the observation that easy data can reduce lr warmup time tell us anything about why easy data can reduce the rewinding time?
- What’s the definition of level in B.2? How much weights are left for each level?

Typos:
L179: “for”, L453: “at least”


**Limitations:**

This paper has some discussion of the limitations in section 6.

**Strengths And Weaknesses:**

Strength:

Finding initializations with sparse trainable networks is a fundamental question in neural network training. The observations made by the paper are interesting and can provide insights into understanding sparse trainable networks. In particular, it shows that using easy data is sufficient and can reduce rewinding time, which can be practically useful.

Weakness:

- The writing of this paper can be improved. I think the connection between the sections can be made stronger. For example, in section 5, the authors found that easy data does not reduce the amount of training needed in the learning rate warmup, but did not talk much about how this phenomenon is related to the previous sections.
- I think this paper could delve deeper into the observations made. While the observations made by the authors are novel, the authors did not give much explanation of the phenomena oberserved. I think the paper can be made much stronger if the authors could propose and investigate some explanations of the phenomena and discuss more how the observations can facilitate new algorithms.

---

> ### Author Response · Authors · 2022-08-02
> **Response to Reviewer xZ2E 1/2**
>
> We thank the reviewer for their helpful comments and suggestions for improving the clarity and content of our paper. While the reviewer found our observations novel and interesting, they identified two main points that needed further clarification: (1) an explanation of the observed phenomena and (2) its connection to the learning rate warmup experiments. We address both below.
>
> **Explanation of why easy data helps.**  Our primary observation is that training on a small fraction of easy data can reduce the number of steps needed to find an initialization which contains a matching sparse network, i.e. one that can be trained to the same accuracy as the dense network. An explanation for this phenomenon can be divided into two questions. First, what characteristic of a pre-trained dense network can predict if it contains a matching sparse network? Second, does training only on easy examples allow us to find networks with this characteristic faster and why?
>
> In Section 4, we seek to provide a linear mode connectivity (LMC) based explanation for the first question. [Previous work](https://arxiv.org/abs/1912.05671) has shown that the rewind step at which sparse networks become matching correlates with the onset of LMC for the *sparse* network. Intuitively, this suggests that a notion of stability is important for finding matching sparse networks—such networks exist at the rewind step at which the *sparse* network optimization becomes stable to SGD noise. However, this occurs before the onset of LMC in the *dense* network. Thus, we generalized the notion of stability to SGD noise from a binary metric—is the train loss barrier below some threshold—to a continuous one—the average train loss barrier—and found that this characteristic of the *dense* network is indeed correlated with our ability to find matching initializations.
>
> To answer the second question, we showed that training only on easy examples finds networks with lower average train loss barriers faster and the resulting distribution of per-example train loss barriers is very similar to those obtained from training on all data for twice as many steps (Section 4, Figure 5). In this revision (Appendix B), we add a new set of experiments to further investigate why this might be the case. First, we compare how fast networks learn when trained on easy vs all data. For CIFAR-100 + ResNet-32, we train for 1600 steps on both a subset of 2048 easy examples and on all examples. We measure the test accuracy as well as the train accuracy on both the full dataset and the easy subset. The results in Figure 7a show that training on easy examples performs better on all 3 metrics, leading to faster learning. Next we compare the coherence of gradients along these training trajectories. Every 10 steps, we sample 16 batches from the dataset used to train the network and calculate the gradients on those batches. For each step, we then measure the cosine similarity between each sampled gradient and the mean gradient across batches. The results in Figure 7b show that this similarity is consistently higher when training on easy data.
>
> [Previous results](https://openreview.net/forum?id=HkxHv4rn24) have shown that networks learn easy examples before learning hard examples. Our results further show that, *very early in training*, making repeated passes through the easy data creates a stronger and more coherent training signal which leads to faster learning. An intuitive summary of our results is, given that the information necessary for finding matching subnetworks is learned in the first few hundred steps, it is more efficient to focus training resources on examples that can be learned well early in training.

---

> > ### Author Response · Authors · 2022-08-09
> > **Any Remaining Questions?**
> >
> > We are eager to address any remaining concerns you have that would make you comfortable recommending acceptance of our paper.  Please let us know if there is anything that we've missed in our rebuttal since we have little time left to make revisions.

---

> > > ### Comment · Reviewer_xZ2E · 2022-08-09
> > > **Response to authors**
> > >
> > > Thanks a lot for your detailed response. I think my concerns have been addressed. I changed my score 5->6.

---

> ### Author Response · Authors · 2022-08-02
> **Response to Reviewer xZ2E 2/2**
>
> **Learning rate warmup.**  The benefits of training on easy data—faster learning and increased stability to SGD noise—are only present in the very early phase of learning. [Previous work](https://arxiv.org/abs/2107.07075) has shown that, if we continue to train on easy data only, the network will converge to a lower final test accuracy. So, a natural question to ask is what other scenarios do the benefits of training on easy data extend to? We chose to investigate this by studying training at high learning rates where [learning can become unstable early in training](https://arxiv.org/abs/2110.04369) and is usually stabilized by learning rate warmup. Our results show that training only on easy examples does not help and in fact hurts performance in this case. Two possible reasons for this are (1) learning rate warmup is often used for the first 10% of training, which may be too long for easy examples to be helpful (the pre-training phase in iterative magnitude pruning where easy examples are clearly beneficial can be as little as 2% of training) and (2) the instability in this case—loss diverging to infinity—is not reduced by methods that improve stability to SGD noise. Despite the negative results of this experiment, we think that, when combined with our previous results, the core scientific findings in our work are interesting and broadly useful to the community. In particular, some **algorithmic implications** are as follows. (1) While developing algorithms for training sparse networks or finding matching networks at initialization, researchers might want to incorporate methods or search for initializations that improve stability to SGD noise. (2) Recent work in [curriculum learning](https://openreview.net/pdf?id=tW4QEInpni) found that curricula provide marginal benefits unless there are additional constraints such as a training budget. Our work helps narrow the time window in which training on easy examples is beneficial and provides another example in which, for the constrained scenario of finding matching sparse networks, a simple curriculum of training on easy data first is beneficial. Thus, these findings may help in the design of better training curricula. (3) In the  training of large language models, a common problem is that loss spikes and training becomes unstable. A hypothesis for why this phenomenon occurs is that the network is presented with a bad batch or a set of bad batches that destabilize training. Since this is related to instability due to SGD noise (bad batches are randomly sampled) it may be possible to improve training stability by injecting easy examples into the training process.
>
>
> **Additional Specific Questions:**
> > B.1 refutes the hypothesis that training on easy data is gradient clipping. Are there any other hypotheses?
>
> See our **explanation of why easy data helps** section above.
>
> > How are the results in section 5 related to finding good sparse initialization? Does the observation that easy data can reduce lr warmup time tell us anything about why easy data can reduce the rewinding time?
>
> See our **learning rate warmup** section for motivation for these experiments.
>
> > What’s the definition of level in B.2? How much weights are left for each level?
>
> Here level refers to the pruning level of Iterative Magnitude Pruning. Throughout the paper 20% of the weights are pruned at each level so Level 0 has 100% of the weights remaining and Level 6 has 26.2% of the weights remaining.
>
> > Typos: L179: “for”, L453: “at least”
>
> Thank you, the typos have been corrected.
>
> **Summary of changes in this revision:**
> * Explanation for why easy data helps find initializations containing matching subnetworks clarified and additional experiments added in Appendix B.
> * Additional motivation for learning rate warmup and algorithmic implications added in Appendix E.
> * Additional experiment with different scores for finding easy data added in Appendix F.1.
> * Currently running additional experiments on different architectures and datasets (VGG-16 on CIFAR-10) and (ResNet-50 on ImageNet-50). These will be added to the camera ready version if accepted. Intermediate results (baseline runs) at: https://anonymous.4open.science/r/36EB/README.md.

---

### Meta-Review · Area_Chair_B1ZK · 2022-08-31

**Recommendation:** Accept
**Confidence:** Certain

**Metareview:**

This paper presents comprehensive experiments studying the role of data in finding lottery tickets in the early stage of training. All reviewers liked the paper and agreed that the paper has novel and insightful results worth sharing with the community.

**Award:**

No

---

### Decision · Program_Chairs · 2022-09-14

Accept